# Frontline Screening for SARS-CoV-2 Infection at Emergency Department Admission by Third Generation Rapid Antigen Test: Can We Spare RT-qPCR?

**DOI:** 10.3390/v13050818

**Published:** 2021-05-01

**Authors:** Valeria Cento, Silvia Renica, Elisa Matarazzo, Maria Antonello, Luna Colagrossi, Federica Di Ruscio, Arianna Pani, Diana Fanti, Chiara Vismara, Massimo Puoti, Francesco Scaglione, Carlo Federico Perno, Claudia Alteri

**Affiliations:** 1Department of Oncology and Hemato-Oncology, Università degli Studi di Milano, 20122 Milan, Italy; valeria.cento@unimi.it (V.C.); silvia.renica@unimi.it (S.R.); maria.antonello1@gmail.com (M.A.); arianna.pani@unimi.it (A.P.); francesco.scaglione@unimi.it (F.S.); 2Residency in Microbiology and Virology, Università degli Studi di Milano, 20122 Milan, Italy; elisa.matarazzo1@unimi.it (E.M.); federica.diruscio@unimi.it (F.D.R.); 3Unit of Microbiology and Diagnostic Immunology, IRCCS Bambino Gesù Children’s Hospital, 00165 Rome, Italy; luna.colagrossi@opbg.net (L.C.); cf.perno@uniroma2.it (C.F.P.); 4Department of Chemical-Clinical and Microbiological Analyses, ASST Grande Ospedale Metropolitano Niguarda, 20162 Milan, Italy; diana.fanti@ospedaleniguarda.it (D.F.); chiara.vismara@ospedaleniguarda.it (C.V.); 5Infectious Diseases Unit, ASST Grande Ospedale Metropolitano Niguarda, 20162 Milan, Italy; massimo.puoti@ospedaleniguarda.it

**Keywords:** antigenic test, SARS-CoV-2, rapid diagnostic test, viral load, subgenomic RNA, infectivity

## Abstract

To complement RT-qPCR testing for diagnosis of severe acute respiratory syndrome coronavirus 2 (SARS-CoV-2) infections, many countries have introduced the use of rapid antigen tests. As they generally display lower real-life performances than expected, their correct positioning as frontline screening is still controversial. Despite the lack of data from daily clinical use, third generation microfluidic assays (such as the LumiraDx SARS-CoV-2 Ag test) have recently been suggested to have similar performances to RT-qPCR and have been proposed as alternative diagnostic tools. By analyzing 960 nasopharyngeal swabs from 960 subjects at the emergency department admissions of a tertiary COVID-19 hospital, LumiraDx assay demonstrated a specificity of 97% (95% CI: 96–98), and a sensitivity of 85% (95% CI: 82–89) in comparison with RT-qPCR, which increases to 91% (95% CI: 86–95) for samples with a cycle threshold ≤ 29. Fifty false-negative LumiraDx-results were confirmed by direct quantification of genomic SARS-CoV-2 RNA through droplet-digital PCR (median (IQR) load = 5880 (1657–41,440) copies/mL). Subgenomic N and E RNAs were detected in 52% (*n* = 26) and 56% (*n* = 28) of them, respectively, supporting the presence of active viral replication. Overall, the LumiraDx test complies with the minimum performance requirements of the WHO. Yet, the risk of a misrecognition of patients with active COVID-19 persists, and the need for confirmatory RT-qPCR should not be amended.

## 1. Introduction

The effective large-scale infection prevention and control strategy of the Severe Acute Respiratory Syndrome Coronavirus 2 (SARS-CoV-2) pandemic relies on a timely and accurate diagnosis of the Coronavirus Disease 2019 (COVID-19), to be able to promptly identify positive cases and halt transmission. The gold standard for this definition is the positivity to real-time polymerase chain reaction (RT-qPCR), providing sensitive and reliable results. However, the high number of daily requests and the concomitant logistical and supply chain issues have led to such an increase in turnaround time as to become incompatible with the needs of an effective large-scale screening strategy. To complement RT-qPCR testing, many countries have introduced in their diagnostic routine the use of rapid antigen tests [1,2,3,4,5,6,7], designed to detect the presence of a viral antigen (usually the nucleocapsid, N). Antigen tests are inexpensive and easy to use, even in point-of-care (POC) scenarios, but with the drawback of a diagnostic sensitivity generally lower than that of RT-qPCR, as they provide reliable results only in the presence of high viral loads (≥100,000 copies/mL), and during the first few days since symptom onset [8,9,10,11,12,13,14]. As a consequence, their correct positioning as frontline screening tests is still controversial.

The WHO, CDC, and ECDC provided specific guidance for the use of rapid antigen test, recommending its use in some settings while underlying the need for a subsequent confirmation of their results by RT-qPCR [8,15,16]. According to the national indications for the case definition of COVID-19 in Italy [3], however, the third generation antigen-detecting tests are considered equivalent to RT-qPCR in terms of performance, and confirmation of their result is not mandatory. At present, however, there are no clinical practice data to support this conclusion, nor the actual ability of these rapid tests to identify potentially contagious individuals.

With this study we aimed to evaluate the concordance between the third generation LumiraDx SARS-CoV-2 Ag test and RT-qPCR at the emergency department (ED) admissions of a tertiary COVID-19 reference hospital. As the antigen test was introduced in our hospital routine to detect and isolate COVID-19 cases within the triage, we focused on the frequency and significance of discordant antigen-negative but RT-qPCR positive results. The direct quantification of SARS-CoV-2 load and subgenomic mRNA (sgmRNA), used as a marker for actively replicating virus [17], was used to investigate the hypothesis that the negativity of the antigen test in presence of a positive RT-PCR result would represent a condition of poor viral replication and, consequently, infectivity.

## 2. Materials and Methods

### 2.1. Study Population

The study was conducted between 21 October 2020 and 9 December 2020 at ASST Grande Ospedale Metropolitano Niguarda. During the selected period, two simultaneous nasopharyngeal swabs (COPAN NP FLOQSwabs^®^; Copan Diagnostics, CA, USA) were collected by trained medical personnel from all subjects who presented themselves at the ED of the hospital with suspected COVID-19 symptoms, or who were in progress to be admitted to hospital care for any other reason. 

All nasopharyngeal swabs were immediately tested upon collection for SARS-CoV-2 by the third generation LumiraDx SARS-CoV-2 Ag test (LumiraDx, UK) and the RT-qPCR (Simplexa™ COVID-19 Direct Kit, DiaSorin; negative threshold ≥ 40 cycles). Residual samples were stored at −80 °C. Demographic and clinical data were obtained retrospectively by pseudonymized electronic medical records by confirmed COVID-19 cases. The severity of the disease was classified in line with the WHO scale [18]. A total of 1146 third-generation antigenic LumiraDx test and 1146 RT-qPCR were performed on coupled nasopharyngeal swabs sampled; 186 samples were excluded due to previous inclusion of the same patient, leaving 960 coupled RT-qPCR and antigenic POC-test results from 960 patients for final analysis (Appendix A).

### 2.2. SARS-CoV-2 Load and Subgenomic RNA Quantification

In discordant cases (positive at RT-qPCR and negative at antigenic LumiraDx test, +/− discordant), SARS-CoV-2 genomic RNA load was assessed by a QX200™ Droplet Digital™ PCR System (ddPCR, Bio-Rad Laboratories, Inc.) on residual nasopharyngeal samples, as described previously [19]. Primers and probes for the detection of subgenomic mRNA (sgmRNA) were designed to include a forward primer targeting the leader sequence, and reverse primers and probes targeting the first 160 nucleotides of the envelope and nucleocapsid coding region of SARS-CoV-2. In particular, SARS-CoV-2 sgmRNA were quantified using ddPCR adapted assays targeting the nucleocapsid (N) sgRNA (Forward: 5′-CCAACCAACTTTCGATCTCTT-3′, Reverse: 5′-GTGAACCAAGACGCAGTATTAT-3′, and FAM Probe: 5′-TGGAGAACGCAG-TGGGGCGCG-3′) and the envelope (E) sgmRNA (Forward: 5′-CGATCTCTT-GTAGATCTGTTCTC-3′, Reverse: 5′-ATATTGCAGCAGTACGCACACA-3′, and HEX Probe: 5′-ACACTAGCCATCCTTACTGCGCTTCG-3′) [17,20]. The cycling conditions were: 45 °C (60 min), 95 °C (10 min), 40 cycles of 95 °C (30 s) and 58 °C (1 min), 98 °C (10 min), and 4 °C (∞). SARS-CoV-2 quantification was finally expressed in number copies/mL of swab.

### 2.3. Statistics

To define the performance of LumiraDx test, sensitivity, specificity, positive predictive value (PPV) and negative predictive value (NPV) were assessed against the RT-qPCR, considered as the gold standard for SARS-CoV-2 testing. A univariate and multivariate logistic regression analyses were also performed in order to define factors associated with the presence of SARS-CoV-2 sgmRNAs used as a marker for actively replicating virus in the subset of discordant cases. Analyses were performed using IBM^®^ SPSS^®^ Statistics version 26 (SPS S.r.l., Bologna, Italy).

## 3. Results

### 3.1. Characteristics of the Study Population

Five hundred fifty-five of the 960 patients enrolled in the study were male (57.8%) with a median age of 66 (interquartile range, IQR: 45–79) years (Table 1). Three hundred forty-seven (36.1%) individuals were determined to be SARS-CoV-2 infected by RT-qPCR testing. Clinical information related to COVID-19 were retrieved for a subset of 176 SARS-CoV-2 positive individuals. Among them, 46 (26.1%) and 13 (7.4%) were affected by a critical and severe form of COVID-19, respectively (Table 1).

### 3.2. POC-Test LumiraDx Assay Performance 

The LumiraDx test detected SARS-CoV-2 in 297/347 SARS-CoV-2 RT-qPCR positive samples (cycle threshold, Ct ORF1ab and S median (IQR) value: 23.1 (18.6–27.2) and 22.1 (17.7–26.1), respectively), while it failed to detect SARS-CoV-2 in 50 RT-qPCR positive samples (Ct ORF1ab and S median (IQR) value: 32.4 (29.6–34.0) and 32.0 (29.0–33.8), respectively), resulting in an overall sensitivity of 85% (95% CI: 82–89) (Table 1 and Table 2). A total of 560/577 samples with a negative RT-qPCR result were negative with LumiraDx, while 17 resulted positive, notwithstanding no evidence of COVID-19-related symptoms, giving an overall specificity of 97% (95% CI: 96%–98%) (Table 1 and Table 2).

According to Ct, the LumiraDx test has a sensitivity of 91% (86–95%) for samples with Ct ≤ 29 (*n* = 139), and of 34% (21–46%) for samples with Ct > 29 (*n* = 59) (Table 2). 

Among the 176 patients whose classification of COVID-19 manifestation was retrieved, LumiraDx failed to detect SARS-CoV-2 in 46 individuals (26.1%) (Table 1). Of note, LumiraDx failed to detect SARS-CoV-2 infection in all individuals affected by mild COVID-19 (14, 100%). The median Ct value of RT-qPCR results was 30.8 (26.1–32.8) for Orf1ab and 31.1 (26.1–32.8) for S, equivalent to 23,625 (2800–199,879) copies/mL (Table 1). 

### 3.3. Viral Load and Subgenomic RNA

As stated before, the LumiraDx test failed to detect SARS-CoV-2 in 50 RT-qPCR-positive samples. Direct quantification of genomic SARS-CoV-2 load by ddPCR revealed the presence of SARS-CoV-2 RNA in all the 50 RT-qPCR-positive/antigen-negative samples, with a median (IQR) load of 5880 (1657–41,440) copies/mL (range: 32.7–1,000,000 copies/mL), confirming the RT-qPCR result (Figure 1). Nine of them had a viral load ≥ 100,000 copies/mL, thus above the threshold that would ensure SARS-CoV-2 detection by LumiraDx. In addition, ddPCR demonstrated the production of sgmRNA for the N-antigen in 26/50 samples (52.0%), and for the E-antigen [17] in 28/50 (56.0%), with a median (IQR) sgmRNA load of 809 (203–7409) copies/mL, and 665 (284–9100) copies/mL, respectively (Figure 2). 

Moreover, in order to define factors associated with the presence of sgmRNA in these 50 RT-qPCR-positive/antigen-negative samples and thus to clarify the role of these replicative intermediates, we performed a multivariate logistic regression analysis shown in Appendix A. By this analysis, the presence of sgmRNAs was dependent on the presence of higher viral loads in nasopharyngeal swabs, but not by the timing of testing after symptom onset (at least in the time frame of 1–10 days), and severity of COVID-19 presentation [18], thus suggesting that these clinical parameters might not be sufficient in discriminating individuals at lower risk of viral infectivity and transmission [21]. Even if the presence of sgmRNAs is dependent on the presence of higher viral loads in nasopharyngeal swabs (Appendix A), sgmRNAs were detected in more than half (56.0%; 23/41) of samples with a genomic RNA load of <100,000 copies/mL, supporting the presence of a potentially replicating and infecting virus even in a subset of samples with lower viral load. 

## 4. Discussion

Thanks to the reduced turnaround time, cost, and expertise required for execution, rapid antigen tests are believed to provide added value in the patient triage process in healthcare settings, or where it is necessary to take prompt public health measures. A number of commercial rapid antigen tests have been licensed to be used in clinical practice [22], but data on their on-field performance are extremely scarce and heterogeneous. Previous studies have indeed reported a broad spectrum of sensitivity for rapid antigen tests when compared to RT-qPCR (22–100%) [10,11,12,14,23,24,25,26]. Owing to the unavoidable heterogeneity of the field studies on rapid antigen tests, their performance results are frequently lower than those reported by the manufacturers [10,11,12,14,23,24,25,26].

To the best of our knowledge, this is the first report on the use of a third generation microfluidic immunofluorescence-based antigen test as a frontline screening tool, in association with RT-qPCR, at ED admission in a real-life hospital setting. Even if the 85% sensitivity here reported is far lower than the 97.6% sensitivity described by the manufacturer [27], the LumiraDX test meets the WHO minimum performance requirements for rapid antigen tests in terms of sensitivity (85% vs. ≥80% recommended), and specificity (97.0% vs. ≥97% recommended) [14], but falls short of the ECDC 90% sensitivity requirement [8]. In a recent comparative study evaluating the performances of four different rapid antigen tests against RT-qPCR and cell culture infectivity on 100 samples collected during a random screening within shared living facilities, the LumiraDx assay demonstrated a sensitivity of 50% [14]. The sensitivity we found in hospital triage screening against RT-qPCR (85.0%) is therefore superior to this estimate, probably due to optimization of the performance setting (ED admission, with a high prevalence of symptomatic subjects), high local incidence of SARS-CoV-2 infection (RT-qPCR positivity rate ≥ 20% over a median (IQR) of 1689 (841–2063) RT-qPCR performed daily), and sample size (960 subjects), in full compliance with WHO and ECDC indications on the use of rapid antigen tests [8,15].

We demonstrated a 7.0% discordance between the LumiraDx antigen and RT-qPCR results, with 5.8% being false-negative results. The direct quantification of SARS-CoV-2 genomic RNA by ddPCR in the 50 samples with discordant antigen-negative and RT-PCR positive results confirmed this false-negative rate. This antigen POC test is able to detect SARS-CoV-2 with a sensitivity of 91% in nasopharyngeal swabs characterized by Ct ≤ 29, but the sensitivity declines substantially when the viral load decreased to Ct values over 30 (34%) and may miss the opportunity to detect later and mild infections. 

Our study also demonstrated that the LumiraDx antigen test may fail to detect SARS-CoV-2 in a number of samples with high levels of nasopharyngeal viral-shedding and evidence of replicative intermediates. The presence of these sgmRNA suggests the effective production of the SARS-CoV-2 proteins, including the N-antigen targeted by LumiraDx test, and provided a proof-of-concept that a negative result of the LumiraDx assay does not exclude the presence of a likely replicating and infecting virus, even in recently symptomatic subjects. Indeed, the role of these sgmRNA has been investigated in several papers, all demonstrating a good agreement between virus culture and sgmRNA detection [17,28,29,30]. Specimens positive for sgmRNA, especially those collected within 8 days of symptom onset, more frequently yield positive virus isolation [17,28,29,30], thus suggesting the role of sgmRNA as a marker of a live, potentially replicating and thus likely transmissible virus [31,32,33,34]. Further research is needed to define if the quantitative sgmRNA estimation (and not only the sgmRNA qualitative detection) may serve as a more accurate marker to predict a viable and transmissible virus.

This study has some limitations. Because of the retrospective nature and size of the included population from hospital clinical practice in a situation of high prevalence of SARS-CoV-2 infection, we could not estimate COVID-19 symptoms, type and duration, and disease severity for all subjects, but only for the 176 RT-PCR confirmed COVID-19 cases. This affects the possibility to assess the efficiency of the antigenic POC test in detecting SARS-CoV-2 in infected but asymptomatic individuals. We are not able to assess the detection efficiency of potential N SARS-CoV-2 variants, or if the discordant results can be affected by N single nucleotide polymorphisms. 

Overall, the LumiraDx test complies with the minimum performance requirements of the WHO and has the potential to perform better than previous generations of rapid antigen tests. This test can also offer some undoubted advantages respect to RT-qPCR, like rapid turnaround time, low cost, and easy use without the need for a microbiological laboratory. All these characteristics make antigenic tests suitable for large-scale mass testing, and for SARS-CoV-2 containment strategies integrating social distancing, masks usage, serial antigen screenings, and contact tracing. However, when used as frontline screening for SARS-CoV-2 infection, the risk of a misrecognition of patients with active COVID-19 still exist, and the need for confirmatory RT-qPCR in symptomatic subjects who test antigen-negative should not be amended.

## Figures and Tables

**Figure 1 viruses-13-00818-f001:**
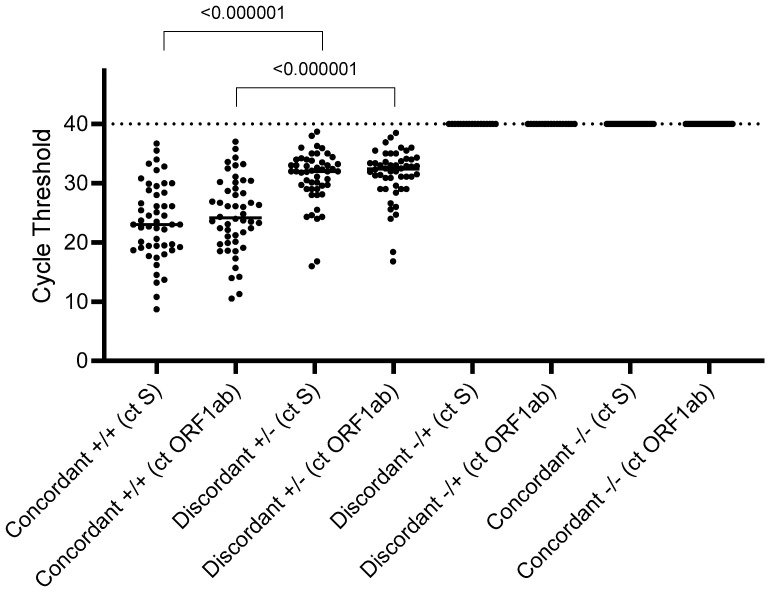
SARS-CoV-2 RT-qPCR cycle thresholds against concordance results between RT-qPCR and LumiraDx SARS-CoV-2 Ag test. One-sided *p*-values comparing Ct of concordant samples (+/+) and Ct of discordant samples (+/−) were calculated by a Mann–Whitney U test. *p*-values < 0.05 were considered statistically significant. Ct = Cycle threshold; ORF1ab = open reading frame 1ab; S = spike.

**Figure 2 viruses-13-00818-f002:**
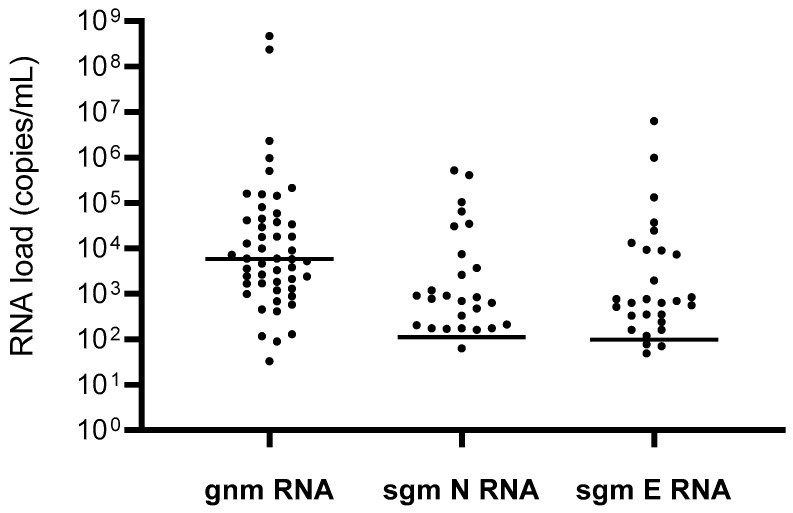
SARS-CoV-2 genomic and subgenomic RNA load in antigenic false-negative samples. Gnm = genomic; sgm = subgenomic; E = envelope; N = nucleocapsid.

**Table 1 viruses-13-00818-t001:** Demographic, clinical, and virological characteristics of the 960 individuals simultaneously screened for SARS-CoV-2 infection by LumiraDx test for the nucleocapsid antigen and RT-qPCR.

	Overall, *n* = 960	Combined Results of RT-qPCR and LumiraDx Test				*p*-Value
		Concordant +/+, *n* = 297	Discordant, +/−, *n* = 50	Concordant −/−, *n* = 596	Discordant, −/+, *n* = 17	
**Demographics and clinical characteristics**						
Age, years	66 (45–79)	66 (51–80)	53 (40–68)	63 (41–79)	60 (46–72)	0.0003
Sex						
*Male*	555 (57.8)	189 (63.6)	37 (74.0)	317 (53.2)	12 (70.6)	0.102
COVID-19-related symptoms ^a^						
*Fever*	104 (59.1)	76 (58.0)	28 (60.9)	-	-	0.458
*Cough*	99 (56.3)	74 (56.9)	25 (54.3)	-	-	0.447
*Dyspnea*	162 (92.0)	130 (100.0)	32 (69.6)	-	-	<0.000001
COVID-19 manifestation ^b^						
*Critical or Severe*	59 (33.5)	43 (33.1)	16 (27.1)	-	-	0.485
*Moderate*	103 (58.5)	87 (66.9)	16 (34.8)	-	-	0.00015
*Mild*	14 (8.0)	0 (0.0)	14 (30.4)	-	-	<0.000001
Time from symptoms onset to SARS-CoV-2 diagnosis ^c^, days	6 (4–7)	5 (4–6)	7 (7–10)	-	-	0.000002
**SARS-CoV-2 RT-qPCR**						
ORF1ab cycle threshold	40 (40–40)	23.1 (18.6–27.2)	32.4 (29.6–34.0)	n.d. 40 (40–40)	n.d. 40 (40–40)	<0.000001
S cycle threshold	40 (40–40)	22.1 (17.7–26.1)	32.0 (29.0–33.8)	n.d. 40 (40–40)	n.d. 40 (40–40)	<0.000001

Data are expressed as median (interquartile range), or N (%). One-sided *p*-values comparing patients with concordant +/+ results and patients with discordant +/− results were calculated by the Mann–Whitney U test, or Fisher’s exact test, as appropriate. *p*-values < 0.05 were considered statistically significant. ^a^ Information available for 176 individuals. ^b^ The severity of the disease was classified in mild, moderate, severe and critical in line with WHO scale. Information available for 176 individuals. ^c^ Information available for 93 individuals. n.d. = not detected.

**Table 2 viruses-13-00818-t002:** Performances of the SARS-CoV-2 antigenic POC test against RT-qPCR results.

	Overall	RT-qPCR SARS-CoV-2 Results	
		Ct ≤ 29	Ct > 29
Sensitivity (CI 95%)	0.85 (0.82–0.89)	0.91 (0.86–0.95)	0.34 (0.21–0.46)
Specificity (CI 95%)	0.97 (0.96–0.98)	-	-
Positive Predictive Value (CI 95%)	0.94 (0.92–0.97)	0.88 (0.83–0.93)	0.52 (0.36–0.69)
Negative Predictive Value (CI 95%)	0.92 (0.90–0.94)	0.98 (0.96–0.99)	0.94 (0.92–0.96)

Sensitivity, specificity, positive predictive value (PPV), and negative predictive value (NPV) were assessed against the RT-qPCR results.

## Data Availability

The data presented in this study are available on request from the corresponding author. The data are not publicly available due to inclusion of clinical and demographic characteristics of patients.

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
