# Peer review of "Frontline Screening for SARS-CoV-2 Infection at Emergency Department Admission by Third Generation Rapid Antigen Test: Can We Spare RT-qPCR?"

_viruses, 2021, doi:10.3390/v13050818_

Round 1
Reviewer 1 Report
In this manuscript titled “Frontline screening for SARS-CoV-2 infection at Emergency Department admission by third generation rapid antigen test: can we spare RT-qPCR?”, Centa et al compared the detection efficiencies of LumiraDxAg test and RT-qPCR. They showed that the rapid antigen test is 85% sensitive compared to the RT-PCR test and propose that the rapid antigen test may be better diagnostic tool at emergency departments with a quick turnaround time.
1.This study lacks novelty, and itonly confirms the sensitivity and specificity claims made by the manufacturer of LumiraDx Antigen test.
2.The authors performed the tests only on patients with typical COVID symptoms such as cough, fever and dyspnea. It would be of more beneficial if they showed that the test can be used to identify asymptomatic carriers of the virus.
3.Further, they suggest that RT-qPCR is still needed. It would be helpful, in the clinical settings, if the authors had figured out a way to achieve equal or better sensitivity of the antigen test compared to the RT-qPCR assays. However, they have not shown that this was achieved.
Author Response
Ref. No.: viruses-1142372
Title: Frontline screening for SARS-CoV-2 infection at Emergency Department admission by third generation rapid antigen test: can we spare RT-qPCR?
Journal: Viruses
Reviewer 1
In this manuscript titled “Frontline screening for SARS-CoV-2 infection at Emergency Department admission by third generation rapid antigen test: can we spare RT-qPCR?”, Cento et al compared the detection efficiencies of LumiraDxAg test and RT-qPCR. They showed that the rapid antigen test is 85% sensitive compared to the RT-PCR test and propose that the rapid antigen test may be better diagnostic tool at emergency departments with a quick turnaround time.
1.This study lacks novelty, and it only confirms the sensitivity and specificity claims made by the manufacturer of LumiraDx Antigen test.
Answer: We agree with the reviewer comment that our results are in line with the sensitivity and specificity reported by the manufacturer of LumiraDx Antigen test, even if the 85% sensitivity reported in our study is far lower than the 97.6% sensitivity described by the manufacturer. Our study also demonstrated that the LumiraDx antigen test may fail to detect SARS-CoV-2 in a number of samples with high level of nasopharyngeal viral-shedding and evidence of replicative-intermediates. In particular, the direct quantification of genomic SARS-CoV-2 load in the 50 Antigen(-)/RT-PCR(+) samples revealed the presence of SARS-CoV-2 RNA in all the samples, with a median (IQR) load of 5,880 (1,657-41,440) copies/mL (range: 32.7-1,000,000 copies/mL. Nine of them had a viral-load ≥100,000 copies/mL, thus above the threshold that would ensure SARS-CoV-2 detection by LumiraDx. In addition, ddPCR demonstrated the production of sgmRNA for the N-antigen in 26/50 samples (52.0%), and for the E-antigen in 28/50 (56.0%), with a median (IQR) sgmRNA load of 809 (203-7,409) copies/mL, and 665 (284-9,100) copies/mL, respectively. The presence of these replicative intermediates suggests the effective production of the proteins, including the N-antigen targeted by LumiraDx test, and provided a proof-of-concept that a negative result of the LumiraDx assay does not exclude the presence of a likely replicating and infecting virus, even in recently symptomatic subjects. This part is reported in results (lines 143-151) and discussion sections (lines 207-213).
2.The authors performed the tests only on patients with typical COVID symptoms such as cough, fever and dyspnea. It would be of more beneficial if they showed that the test can be used to identify asymptomatic carriers of the virus.
Answer: We agree with the limitation highlighted by this reviewer and reviewer 5. For this reason, this limitation was added in the discussion section as follows: Because of the retrospective nature and size of the included population from hospital clinical practice in a situation of high prevalence of SARS-CoV-2 infection, we could not estimate COVID-19 symptoms, type and duration, and disease severity for all subjects, but only for the 176 RT-PCR confirmed COVID-19 cases. This affects the possibility to assess the efficiency of the antigenic POC-test in detecting SARS-CoV-2 in infected but asymptomatic individuals (Discussion, lines 221-228).
3.Further, they suggest that RT-qPCR is still needed. It would be helpful, in the clinical settings, if the authors had figured out a way to achieve equal or better sensitivity of the antigen test compared to the RT-qPCR assays. However, they have not shown that this was achieved.
Answer: In order to better describe our findings, and the utility of antigenic tests in epidemiological setting, conclusion was implemented as follows (Discussion, lines 229-238): Our results suggest that LumiraDx test complies with the minimum performance requirements of the WHO, and has the potential to perform better than previous generations of rapid antigen tests. This test can also offer some undoubted advantages respect to RT-qPCR, like the rapid turnaround time, the low cost and the easy use without the need for a microbiological laboratory. All these characteristics make antigenic tests suitable for large scale-mass testing, and for SARS-CoV-2 containment strategies integrating social distance, masks usage, serial antigen screenings, and contact tracing. Yet, whether used as frontline screening for SARS-CoV-2 infection, the risk of a misrecognition of patients with active COVID-19 still exist, and the need for confirmatory RT-qPCR in symptomatic subjects who test antigen-negative should not be amended.
Reviewer 2 Report
The manuscript by Cento et al., was a relatively comprehensive research article on RNA-based test for the SARS-CoV-2 viruses. The authors tested the potential PCR products and compared the methodology. The whole research might be helpful for understanding of identification of the disease from clinical samples. Experiments were well designed and data was well collected. There were no significant concerns.
Author Response
Ref. No.: viruses-1142372
Title: Frontline screening for SARS-CoV-2 infection at Emergency Department admission by third generation rapid antigen test: can we spare RT-qPCR?
Journal: Viruses
Reviewer 2
The manuscript by Cento et al., was a relatively comprehensive research article on RNA-based test for the SARS-CoV-2 viruses. The authors tested the potential PCR products and compared the methodology. The whole research might be helpful for understanding of identification of the disease from clinical samples. Experiments were well designed and data was well collected. There were no significant concerns.
Answer: We would like to thank the reviewer for his/her time and comments.
Reviewer 3 Report
Valeria Cento and authors put together a very nice study and manuscript analyzing the LumiraDx SARS-CoV-2 testing platform. I have no major concerns but would like to suggest to the authors they expand the discussion to include two items. The first, unless I missed it I don't see much discussing time as an advantage for the LumiraDx. I also think a discussion of cost would be worthwhile. Otherwise, the discussion and conclusions were a fair assessment of the technology as compared to qRT-PCR as a diagnostic tool.
Author Response
Ref. No.: viruses-1142372
Title: Frontline screening for SARS-CoV-2 infection at Emergency Department admission by third generation rapid antigen test: can we spare RT-qPCR?
Journal: Viruses
Reviewer 3
Valeria Cento and authors put together a very nice study and manuscript analyzing the LumiraDx SARS-CoV-2 testing platform. I have no major concerns but would like to suggest to the authors they expand the discussion to include two items. The first, unless I missed it I don't see much discussing time as an advantage for the LumiraDx. I also think a discussion of cost would be worthwhile. Otherwise, the discussion and conclusions were a fair assessment of the technology as compared to qRT-PCR as a diagnostic tool.
Answer: We thank the reviewer for her/his comment.
According with the revision requested, discussion was overall revised, and the conclusion was implemented as follows (lines 229-238):
Overall, the LumiraDx test complies with the minimum performance requirements of the WHO, and has the potential to perform better than previous generations of rapid antigen tests. This test can also offer some undoubted advantages respect to RT-qPCR, like the rapid turnaround time, the low cost and the easy use without the need for a microbiological laboratory. All these characteristics make antigenic tests suitable for large scale-mass testing, and for SARS-CoV-2 containment strategies integrating social distance, masks usage, serial antigen screenings, and contact tracing. Yet, whether used as frontline screening for SARS-CoV-2 infection, the risk of a misrecognition of patients with active COVID-19 still exist, and the need for confirmatory RT-qPCR in symptomatic subjects who test antigen-negative should not be amended.
Reviewer 4 Report
Excellent work and a very relevant paper adding to our understanding of the limitations of Later flow devices when compared to the gold standard PCR tests.
The authors should provide a bit more information on the in-house method they used to measure subgenomic RNA and also to explain how these measurements correlated with viral culture results where these studies exist. What are the limitations with measuring sgmRNA?. It would have been interesting to have done sgmRNA levels on all the samples, but appreciate that may not have been possible.
Not sure the supplemental Table adds much to the paper
I fully agree with the conclusions drawn.
Author Response
Ref. No.: viruses-1142372
Title: Frontline screening for SARS-CoV-2 infection at Emergency Department admission by third generation rapid antigen test: can we spare RT-qPCR?
Journal: Viruses
Reviewer 4
Excellent work and a very relevant paper adding to our understanding of the limitations of Later flow devices when compared to the gold standard PCR tests.
Answer: We would like to thank the reviewer for his/her comment.
- The authors should provide a bit more information on the in-house method they used to measure subgenomic RNA and also to explain how these measurements correlated with viral culture results where these studies exist. What are the limitations with measuring sgmRNA? It would have been interesting to have done sgmRNA levels on all the samples, but appreciate that may not have been possible.
Answer: Primers and probes for the detection of subgenomic mRNA (sgmRNA) were designed to include a forward primer targeting the leader sequence, and reverse primers and probes targeting the first 160 nucleotides of the envelope and nucleocapsid coding region of SARS-CoV-2. In particular, SARS-CoV-2 sgmRNA were quantified using ddPCR adapted assays targeting the Nucleocapsid (N) sgRNA (Forward: 5’- CCAACCAACTTTCGATCTCTT-3’, Reverse: 5’- GTGAACCAAGACGCAGTATTAT -3’ and FAM Probe: 5’- TGGAGAACGCAG-TGGGGCGCG -3’) and the Envelope (E) sgmRNA (Forward: 5’- CGATCTCTT-GTAGATCTGTTCTC -3’, Reverse: 5’- ATATTGCAGCAGTACGCACACA -3’ and HEX Probe: 5’- ACACTAGCCATCCTTACTGCGCTTCG -3’). By this analysis, we demonstrated that the LumiraDx antigen test may fail to detect SARS-CoV-2 in a number of samples with high level of nasopharyngeal viral-shedding and evidence of replicative-intermediates. The presence of these sgmRNA suggests the effective production of the SARS-CoV-2 proteins, including the N-antigen targeted by LumiraDx test, and provided a proof-of-concept that a negative result of the LumiraDx assay does not exclude the presence of a likely replicating and infecting virus, even in recently symptomatic subjects. The role of these sgmRNA has been investigated in several papers, all demonstrating a good agreement between virus culture and sgmRNA detection (Perera RPM et al., Emer Inf Dis 2021; Ford L et al., CID 2021). Specimens positive for sgmRNA, especially those collected within 8 days of symptom onset, more frequently yield positive virus isolation (Wolfel et al., Nature 2020, La Scola B et al, E J Clin Micro & Inf Dis 2020, Perera RPM et al., Emer Inf Dis 2021), thus suggesting the role of sgmRNA as a marker of live, potentially replicating and thus likely transmissible virus (Corbett KS et al, NEJM 2020; van Doremalen et al., Nature 2020; Yu J et al., Science 2020; Sola I et al., Ann Rev Virol 2020). Further research is needed to define if the quantitative sgmRNA estimation (and not only the sgmRNA qualitative detection) may serve as more accurate marker to predict viable and transmissible virus.
This information is now reported in the revised version of the manuscript (Methods, lines 88-97; Discussion, lines 207-220). Primers and probes of N sgmRNA have been modified due to a mistake done in the previous version of the manuscript. We apologize for this error.
- Not sure the supplemental Table adds much to the paper
Answer: In order to define factors associated with the presence of sgmRNA in the 50 RT-qPCR-positive/antigen-negative samples and thus to clarify the role of these replicative intermediates, we performed a multivariate logistic regression analysis shown in Supplementary Table 1. By this analysis, the presence of sgmRNAs was dependent by the presence of higher viral loads in nasopharyngeal swabs, but not by the timing of testing after symptoms onset (at least in the time frame of 1-10 days), and severity of COVID-19 presentation, thus suggesting that these clinical parameters might be not sufficient in discriminating individuals at lower risk of viral infectivity and transmission. This concept is better clarified in the results section (lines 152-164).
- I fully agree with the conclusions drawn.
Answer: We would like to thank the reviewer for his/her comment.
Reviewer 5 Report
Dear Authors,
thank you very much for the possibility to review your manuscript titled:" Frontline screening for SARS-CoV-2 infection at Emergency Department admission by third generation rapid antigen test: can we spare RT-qPCR?"
The authors present a diagnostic study comparing performance of 3rd Generation Ag Tests versus RT-qPCR results for SARS Cov 2. Overall 960 patients presenting to a emergency department of a tertiary care hospital in Italy being either symptomatic or asymptomatic for SARS CoV2 were tested with nasopharyngeal swabs.
The authors find that the Ag test evaluated in their study fulfils diagnostic criteria by the WHO, but has a inferior sensitivity compared to RT-qPCR.
If possible I would like to suggest a few minor points, which could improve the submission in my opinion.
Please describe where, who and how did you perform the Ag test in the the methods section
Table 1. please Change time from onset of symptoms to diagnosis from weeks to days
Table 2. Please provide additional tables in the supplement for asymptomatic and symptomatic patients.
Author Response
Ref. No.: viruses-1142372
Title: Frontline screening for SARS-CoV-2 infection at Emergency Department admission by third generation rapid antigen test: can we spare RT-qPCR?
Journal: Viruses
Reviewer 5
Dear Authors,
thank you very much for the possibility to review your manuscript titled:" Frontline screening for SARS-CoV-2 infection at Emergency Department admission by third generation rapid antigen test: can we spare RT-qPCR?" The authors present a diagnostic study comparing performance of 3rd Generation Ag Tests versus RT-qPCR results for SARS Cov 2. Overall 960 patients presenting to a emergency department of a tertiary care hospital in Italy, being either symptomatic or asymptomatic for SARS CoV2 were tested with nasopharyngeal swabs. The authors find that the Ag test evaluated in their study fulfils diagnostic criteria by the WHO, but has a inferior sensitivity compared to RT-qPCR. If possible I would like to suggest a few minor points, which could improve the submission in my opinion.
Answer: We would like to thank the reviewer for his/her comment.
- Please describe where, who and how did you perform the Ag test in the methods section.
Answer: We moved this part (extensively revised) from the results section to the methods section, accordingly (lines 69-83).
- Table 1. please Change time from onset of symptoms to diagnosis from weeks to days.
Answer: Done.
- Table 2. Please provide additional tables in the supplement for asymptomatic and symptomatic patients.
Answer: Unfortunately, because of the retrospective nature and size of the included population from hospital clinical practice in a situation of high prevalence of SARS-CoV-2 infection, we could not estimate COVID-19 symptoms, type and duration, and disease severity for all subjects, but only for the 176 RT-PCR confirmed COVID-19 cases. This affects the possibility to assess the efficiency of the antigenic POC-test in detecting SARS-CoV-2 in infected but asymptomatic individuals. This is now reported in the discussion section (lines 221-228).
Round 2
Reviewer 1 Report
The authors thoroughly addressed the comments by expanding the discussion.